

# Multi-hazards in Scandinavia: Impacts and risks from compound heatwaves, droughts and wildfires

Gwendoline Ducros[3], Timothy Tiggeloven[1], Lin Ma[2], Anne Sophie Daloz[2], Nina Schuhen[2], and Marleen C. de Ruiter[1]

[1] *Institute for Environmental Studies, Vrije Universitet Amsterdam, 1081 HV Amsterdam, The Netherlands*

[2] *CICERO Center for International Climate Research, Oslo 0318, Norway*

[3] *independent researcher (formerly at: Institute for Environmental Studies, Vrije Universitet Amsterdam, 1081 HV Amsterdam, The Netherlands)*

*Correspondence to:* Gwendoline Ducros (gwendolineducros@gmail.com)

**Abstract.**

In the summer of 2018, large parts of Scandinavia faced record-breaking heat and drought, leading to increased mortality, agricultural water shortages, hydropower deficits, and higher energy prices. The 2018 heatwave coupled with droughts leading to wildfires are described as multi-hazard events, defined as compounding, cascading or consecutive events. Climate change is driving an increase in heat-related events and, subsequently, shows the necessity to prepare for such hazards, and to assess suitable adaptation measures. To better understand the interplay of multi-hazard risk of heatwaves, droughts and wildfires in a multi-sectoral context and to improve disaster risk management in a multi-hazard setting, we assess the occurrence of these hazards using a spatial analysis of compound heatwave, drought and wildfire events in Scandinavia. To assess their potential direct and indirect economic impacts we use the global Computable General Equilibrium (CGE) model GRACE (Global Responses to Anthropogenic Changes in the Environment) and the 2018 heatwave-drought period as a baseline to map multi-hazard risk. We find that multi-hazard events are pronounced in the summer months in Scandinavia and the 2018 multi-hazard events did not occur in isolation. The 2018 multi-hazard events led to a 0.08% GDP drop in Scandinavia, with forestry experiencing a 3.04% decline, affecting agriculture, electricity, and forestry exports, which dropped by 29.39%, impacting Europe's trade balance. This research shows the importance of ripple effects of multi-hazard, and that forest management and adaptation measures are vital to reducing the risks of heat-related multi-hazards in vulnerable areas.



## 1 Introduction

In the summer of 2018, in particular over the period May-August, large parts of Scandinavia experienced record-breaking temperatures and extreme drought (Bakke et al., 2020). These climate conditions were linked to severe repercussions on human health and the ecosystem, leading to an increased mortality rate during that period (Åström et al., 2019), water shortages that impacted agricultural areas (Buras et al., 2020), as well as hydropower energy deficit and an increase in energy prices (Norwegian Water Resources and Energy Directorate, 2018). The temperature anomalies experienced during the months of May to July were found to be enhanced by human-induced climate change (Wilcke et al., 2020), amongst other factors (Kueh et Lin, 2020).

Heat-related events are expected to increase in frequency, severity, and intensity in the future as a result of anthropogenic climate change (IPCC, 2021). Anthropogenic climate change is also predicted to intensify fire and drought frequency in boreal ecosystems (Girardin et al., 2010; IPCC, 2021) with winter warming expected to increase in boreal forests due to decreasing snow cover and albedo (IPCC, 2021). Spatial patterns of snow cover already show a declining trend in Scandinavia (Brown and Mote, 2009) and the northern area of Scandinavia even sees a projected increase in temperature twice as much as average global warming in winter (Christensen et al., 2022).

The 2018 heatwave coupled with droughts leading to wildfires events are described as multi-hazard events which can occur as compound events if they happen simultaneously, or consecutive events if they occur one after the other (Sutanto et al., 2020; Zscheischler et al., 2017; De Ruiter et al., 2020). This study will focus on compound events, defined here as two or more extreme events occurring at the same time (same day and same region), following the definition from Zscheischler et al., 2017. Specific compound events can be explained by feedback mechanisms, where interactions between climate processes can lead to a positive feedback loop and exacerbate the effects of multiple hazards (IPCC 2012; Zscheischler et al., 2017; Raymond et al., 2020; AghaKouchak et al., 2020). Tilloy et al. (2019) provided a thorough overview of different quantification methods used in the literature for multi-hazard interactions, classifying approaches in stochastic, empirical, and mechanistic methods. In recent years, compound studies have increasingly made use of multivariate-statistical modeling techniques (Couasnon et al. 2020; Mazdiyasni & AghaKouchak 2015; Paprotny et al. 2020; Moftakhari et al. 2019; Wahl et al. 2015).

The projected increase in heat-related events shows the necessity to prepare for such hazards, and to assess suitable adaptation measures. Although the probability of compound events is predicted to increase with the rise in global temperature (IPCC, 2021), mitigation and adaptation measures for multi-hazard compound events have only recently begun to be addressed. Frameworks such as the Sendai Framework for Disaster Risk Reduction (SFDRR) have been adopted by the United Nations with the goal of decreasing disaster risk and increasing resilience, underlining the importance of looking at multi-hazard risk when implementing Disaster Risk Reduction (DRR) measures (UNDRR, 2015). Several studies have emphasized that adaptation strategies and policies are more effective when taking into account multiple stressors (Scolobig et al., 2017; IPCC 2012; de Ruiter et al. 2021; Schipper, 2020; Berrang-Ford et al., 2021). Research has found that certain adaptation measures put into place for a specific hazard might negatively impact adaptation measures against another hazard (de Ruiter et al., 2021), such as the potential of flood DRR measures to increase the risk of droughts and vice versa (Ward et al., 2020). Accounting for multi-hazards in DRR measures decreases the probability that an adaptation measure designed for a singular hazard increases the risk for another (Zscheischler et al., 2017; Raymond et al., 2020; AghaKouchak et al., 2020).

Moreover, heat-related events can have severe direct and indirect economic impacts on sectors such as agricultural or energy production. For example, annual economic losses caused by droughts are currently estimated at around 9€ billion for the EU and the UK, with agriculture amounting to 30-60% of losses and the energy sector to 22-48% (European Commission: Joint Research Centre, 2020). Nearing the end of the 21st century, these losses are estimated between 25 and 45€ billion, depending on the climate scenario, and with no adaptations put into place (European Commission: Joint Research Centre, 2020). Additionally, socio-economic impacts of compound events may surpass those predicted by examining each driver individually (Matano et al., 2021). With this perspective it is thus crucial to include multi-hazard risk when analyzing economic impacts of heat-related events.

To better understand the interplay of multi-hazard risk of heatwaves, droughts and wildfires in a multi-sectoral context and to improve disaster risk management in a multi-hazard setting, we assess the occurrence of these



hazards using a spatial analysis of compound drought, wildfire, and heatwave events from 2000 to 2018 in Scandinavia
(here Finland, Norway and Sweden). To assess their potential direct and indirect economic impacts we use the global
Computable general equilibrium (CGE) model GRACE (Global Responses to Anthropogenic Changes in the
Environment) and the 2018 heatwave-drought period as a baseline. CGE models or partial equilibrium models are
commonly used to evaluate the economic impacts of changes in agriculture and food production (Ntombela et al.,
2017, Manuel et al., 2021; Solomon et al., 2021).

**2 Data & Methods**
To assess and better understand the multi-hazard risk and impacts in Scandinavia during the 2018 multi-hazard event,
we first identify past trends and patterns, as these provide essential context for evaluating the event's economic
impacts. Our methodology includes the following steps: First, we define historical multi-hazard events using the
ERA5 global climate and weather reanalysis (Hersbach et al., 2023) and a copula function describing the correlation
structure between key variables (section 2.1). The second part (section 2.2) focuses on mapping multi-hazard risk, for
which we will map different combinations of compound heatwave, droughts and wildfires events over the period
2000-2018, and, from this, map the 90th percentile of the compound multi-hazards. Section 2.3 maps out Scandinavian
land cover using data from the Copernicus Land Monitoring Service, and uses the multi-hazard risk maps and land
cover maps to map the main land cover types at highest risk of multi-hazards. Lastly, section 2.4 will cover the
economic impacts of the 2018 multi-hazard event using GRACE. Together, these methods increase understanding of
Scandinavian multi-hazard events in summer.

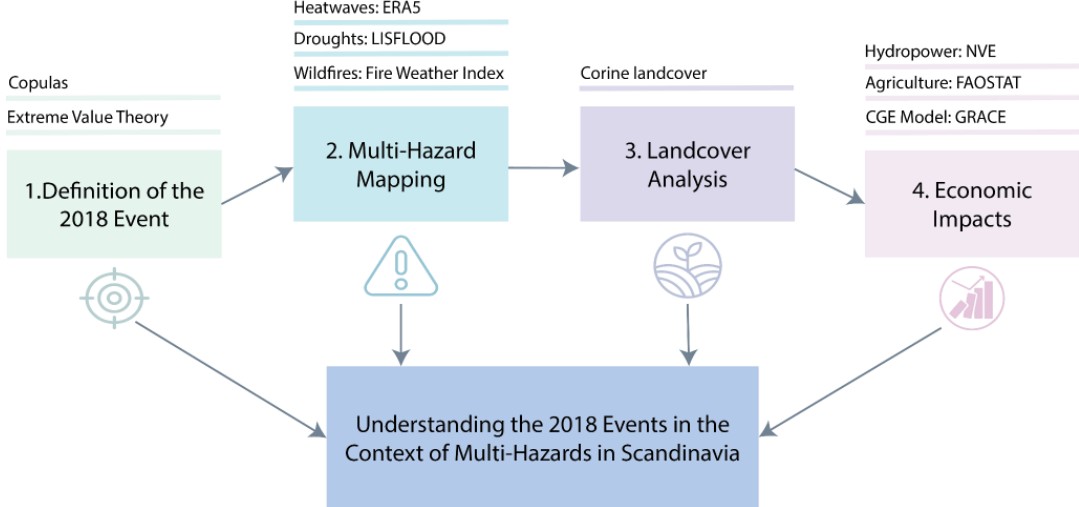

**Figure 1.** Flowchart describing the methodology and data used in the study
**2.1 Definition of the 2018 multi-hazard event**
We investigate the optimal objective definition of the heat wave and drought compound event that occurred across
Europe in spring and summer 2018. As the area of interest is restricted to Northern European countries - namely
Finland, Norway and Sweden - the domain is restricted to land masses of these three countries (below 67˚ N). The
analysis is based on data from the ERA5 reanalysis for March to September, in the time period 1979-2023 on daily
temporal resolution.
To obtain the event definition, we proceed in three steps, first estimating climatological distributions for each
variable (daily maximum surface temperature and total precipitation) from the full data set, then connecting the



univariate distributions with a copula to get a multivariate joint distribution, and finally look for the time period with
the smallest event probability, based on the maxime that the extreme nature of such an event is best characterized by
minimizing its rarity (Schuhen et al. (2024); see Cattiaux and Ribes (2018) for more details on the univariate
procedure).
To estimate the marginal distributions, we collect, for each day and year in March-September of the data set,
the maximum (for temperature) or minimum (for precipitation) value over a temporal neighborhood of 7 days on
either side. This is to ensure a more robust and smoother estimation. For a large range of potential event dates and
durations, these values are then averaged over the respective time period and area, before a standard probability
distribution is fitted to each scale (Gaussian distributions for temperature and generalized extreme value distributions
for precipitation).
To combine the two marginal distributions into a bivariate distribution, we use a copula, which is a
multivariate cumulative distribution function describing the correlation structure between the variables, independent
of the marginal distributions. In this case, we found that the symmetrical Frank copula best represents the relationship
between temperature and precipitation. Finally, we compute from this distribution the joint probability that
temperature would exceed the 2018 value and precipitation would be lower than the 2018 values. This procedure is
repeated for the whole range of potential temporal scales of different dates and durations between May and September
2018, and we finally find the minimum in the set of probabilities, which is associated with the objective event
definition.
**2.2 Multi-hazard mapping of historical events**
This study builds on and uses datasets previously generated by Sutanto et al., (2020) containing the required heatwave,
drought and wildfire data that have been used for this analysis. Sutanto et al. (2020) analyzed drought, heatwave and
wildfire events occurring in the months of June, July and August (JJA) from 1990 to 2018 at the pan-European scale.
Weather data for heat waves was drawn from ERA5, soil moisture drought simulated through the LISFLOOD model,
and wildfire estimated with the Fire Weather Index. They analyzed the frequency and spatial distribution of
occurrences of these hazards, and created daily binary maps (0 indicating no risk, and 1 indicating a risk). This resulted
in three datasets of 2886 maps each (one map for each summer day of JJA over the period 1990-2018).
The Copernicus Land Monitoring Service inventories Scandinavian land cover starting in the year 2000. For
consistency with the Corine Land Cover (CLC) datasets that are used in part 2.2, the study period for this research is
thus 2000-2018. The hazard datasets were analyzed to create four compound hazards maps of the following
combinations: drought and wildfire (DF), heatwave and wildfire (HF), heatwave and drought (HD), and drought,
heatwave and wildfire (DHF) over the period 2000-2018. First, we developed maps indicating the percent of summer
days at risk of the hazard combinations by adding for each hazard combination, the individual hazard maps together,
and dividing by the amount of summer days over the period 1990-2018. To cover the study period, a subset was
created from the period 1990-2018 to cover the period 2000-2018, for each hazard combination, and divided by the
amount of summer days during the 2000-2018 period, which corresponds to 1656 summer days (see formula below).
The resulting maps indicated the percent of summer days at risk of each hazard combination.
*Percent of days with an event*
$= (number\ of\ days\ with\ an\ event\ /\ total\ number\ of\ summer\ days) * 100$
To simplify the compound hazard maps, we calculated the 90th percentile of percent of days of each
compound hazard map to produce binary compound risk maps, with 0 corresponding to no risk of a compound hazard
combination, and 1 to a risk of a compound hazard combination. The above-mentioned percentile was chosen
following Sutanto et al.'s (2020) calculations.
Lastly, compound hazard maps for the study area, Scandinavia, were generated by clipping the binary 90th
percentile compound hazard maps with the region of Scandinavia, from Nuts-1 region maps provided by Eurostat, and
extracting only the Scandinavian region.



**2.3 Analysis of land cover type in areas at high risk of multi-hazard compound events**

Next, we generated a land cover map of Scandinavia using data from the CORINE Land Cover (CLC) inventory provided by the Copernicus Land Monitoring Service. The raster files over the period 2000-2018 and CLC legend were used to classify land cover types, as seen in annex 1. Figure 4 below shows the percentage share of land cover in Scandinavia for the year 2018.

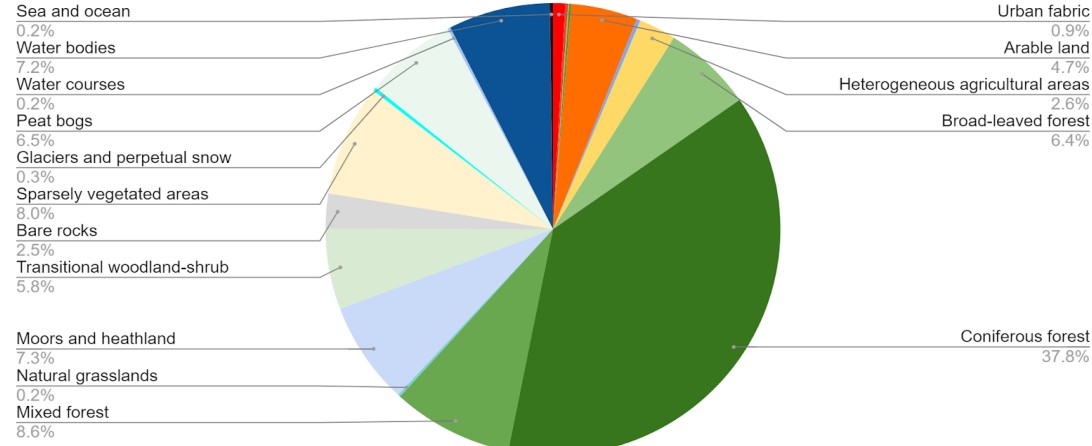

**Figure 2.** Percent of total land per land cover type in 2018.

As seen in Fig. 2, coniferous forests cover the majority of Scandinavia, accounting for 37.8% of the total surface area and mostly in Sweden and Finland where land cover is largely dominated by coniferous forests, as expected due to their economic reliance on the timber industry. Mixed forests and sparsely vegetated areas are the next most extensive land cover, accounting for 8.6% and 8.0% of the total surface area, respectively. In contrast to Sweden and Finland, Norway has a very high proportion of this sparsely vegetated area, and broad-leaved forests (6.4% of Scandinavia) are also mainly found in Norway and along the Norwegian-Swedish border. Arable land and heterogeneous agricultural areas account for respectively 4.7% and 2.6% of total land cover. Urban fabric only accounts for 0.9% of the total surface area in the region whilst the vast majority of arable land is located in Southern Sweden and Finland.

In order to produce the main land cover types affected by the studied compound hazard combinations, the multi-hazard maps of Scandinavia from section 2.2.2 were overlaid with the land cover map of Scandinavia generated in 2.2.3, using the multi-hazard maps as references for resolution. Subsequently, we derived the amount of land affected by each hazard combination per land cover type with spatial analysis.

**2.4 Assessing economic impacts of multi-hazard risk in Scandinavia**

We define here direct economic loss as "monetary value of total or partial destruction of physical assets existing in the affected area" and indirect economic loss as "a decline in economic value added as a consequence of direct economic loss and/or human and environmental impacts" (UNDRR, 2017).

**2.4.1 Economic Model**

To assess the economic impacts of multi-hazard risk, we employ a multi-region, multi-sector computable general equilibrium (CGE) model, Global Responses to Anthropogenic Changes in the Environment (GRACE) (Aaheim et al.





2018). The GRACE model follows the standard assumptions in most CGE models, including assumptions for
producers, Regional Households (RH) and the market dynamics. In this paper, the parameters in the GRACE model
are calibrated using the global social accounting data in 2014 in the Global Trade Analysis Project (GTAP) database
version 10 (Aguiar et al., 2019). In order to address the impacts of the 2018 Scandinavia multi-hazards within the
European area, we divided the global region into 33 European countries[1] and the rest of the world. Each country and
region is further divided into 11 sectors: agriculture, forestry, fishery, manufacturing, services, transportation, crude
oil, coal, gas, refinery, and electricity. The static version of the GRACE model is solved at the country-sectoral level
on an annual basis. With significant advantages in the GRACE model due to the multi-sector and multi-region setup,
the model is able to provide a comprehensive analysis on how the sectoral-specific shocks, such as those caused by
the natural hazards, transfer to other sectors and parts of the economy through the value chain effects. Meanwhile, it
also reveals how the country-specific effects due to hazards spill over to other regions through the trade, which makes
it particularly useful for assessing the broader economic consequence of multi-hazard events.

### 2.4.2 Sectoral context in Scandinavia

As discussed in the previous section, the direct impacts of the 2018 multi-hazards are mainly focused on agriculture,
forestry, and energy. Therefore, in this research, we employ various methods to assess the direct physical impact of
2018 natural hazards on the production of these targeted sectors, which is the input of the macroeconomic model for
evaluating the indirect impacts.

### 2.4.3 Estimating sectoral heat-induced impact

For estimating the direct sectoral impact functions, we employed the dataset on the annual production of agriculture
goods in the Scandinavia region for the period 1961 – 2020 from Food and Agriculture Organization Corporate
Statistical Database (FAOSTAT) (2023). This study approximates total hydroelectricity production using aggregate
reservoir storage volume, as recommended by Norwegian Water Resources and Energy Directorate (NVE) (2019).
Due to the limited availability of daily hydroelectricity production data, we use the weekly reservoir level for Norway
for the period 1995–2022, collected from NVE (2024), as the representative of the region. This estimation employs
climate data extracted from Lund et al. (2023)

We employ econometric models to assess the direct physical impacts of extreme weather events on agriculture and
energy production following Aaheim et al. (2012). Initially, we estimate the relationship between climatic variables
and sector-specific outputs, utilizing the observational data detailed in Section 2.3.2. For this analysis, a log-level
model is employed. The model formulation is as follows:

$$log\left(Q_t^i\right) = \alpha_i + \beta_i X_t^{climate} + \pi_i X_t^{control},$$

where $Q_t^i$ represents the production of sector $i$. $X_t^{climate}$ denotes the vector of climate variables, which includes the
average weekly temperature, precipitation and their interaction terms. It includes [ $\Delta T, \Delta P, T \times \Delta T, P \times$
$\Delta P, \Delta T^2, \Delta P^2, T \times P$ ]. $X_t^{control}$ denotes the vector of control variables. When estimating the impact function for the
energy sector, $X_t^{control}$ comprises month, year, and country dummy variables. When estimating the impact function
for the agriculture and forestry sectors, $X_t^{control}$ includes year and country dummy variables.
To estimate the impact functions, we utilize the forward selection method. This stepwise regression approach
identifies the most significant variables for inclusion in our regression model. We selected the regression model that
best fits the empirical data, as indicated by the highest R-squared value. Table 1 reports the estimated percentage
change of production of agriculture and electricity products, $\beta_i$, in the climate-impact functions. Only estimates that
are statistically significant at confidence level α=0.05 are reported and employed in the GRACE model. All values

---

[1] The European countries are Austria, Belgium, Bulgaria, Croatia, Cyprus, Czech Republic, Denmark, Estonia, Finland, France, Germany, Greece, Hungary, Ireland, Italy, Latvia, Lithuania, Luxembourg, Malta, Netherlands, Poland, Portugal, Romania, Slovakia, Slovenia, Spain, Sweden, United Kingdom, Switzerland, Norway, Albania, Belarus, Ukraine.





have been adjusted to annualized measures for consistency used for the assessment of economic impacts in the
GRACE. These results update the previous estimation outcome in Aaheim et al. (2012) for the Scandinavia region,
and the magnitude of unit impacts remains consistent.
**Table 1.** Estimated percentage change of sectoral production (annualized)

| SECTOR | $\Delta T$ | $\Delta P$ | $T \times \Delta T$ | $\Delta P^2$ | $\Delta T^2$ |
|---|---|---|---|---|---|
| Agriculture | 0.0045 | 0.0123 | -0.0012 | 0.0007 | 0.0014 |
| Electricity | 0.0076 | 0.0035 | | | -0.0007 |

Next, we assess the direct physical impact on agricultural and energy production resulting from extreme weather
events in 2018. To do this, we calculate the 95th percentile of climate variable deviations from their climatological
norms for the year 2018. Our analysis reveals a significant deviation, indicating a temperature increase of 5.50℃ and
a precipitation decrease of 170 mm.
Finally, we assess the impact on the forestry sector. We utilized the assessment detailed in Section 3.2, and
computed 6% of the forest area was affected by the multi-hazard event. We use the share of the area impacted by the
drought-wildfire-heat events as a proxy to assess the effect on the production of forestry. However, this approach
oversimplifies by not accounting for the heterogeneity of plant density and yield rates across different tree species.
This could potentially lead to inaccuracies in our measurements, which could be extended for further research.
**3 Results**
**3.1 Spatial distribution of compound hazard events in Scandinavia**
High risk drought-wildfire events occur twice as often as heat-wildfire, and heat-drought events, with occurrences up
to 166 days of the summer seasons between 2000-2018, as seen on Fig. 3 below. The majority of areas across
Scandinavia have low risk of compound combination hazards (from 0 to 41 days of the summer seasons). Although
the risk is currently very low for most areas, droughts and wildfires in boreal ecosystems are expected to escalate with
rising global temperatures (IPCC, 2021).





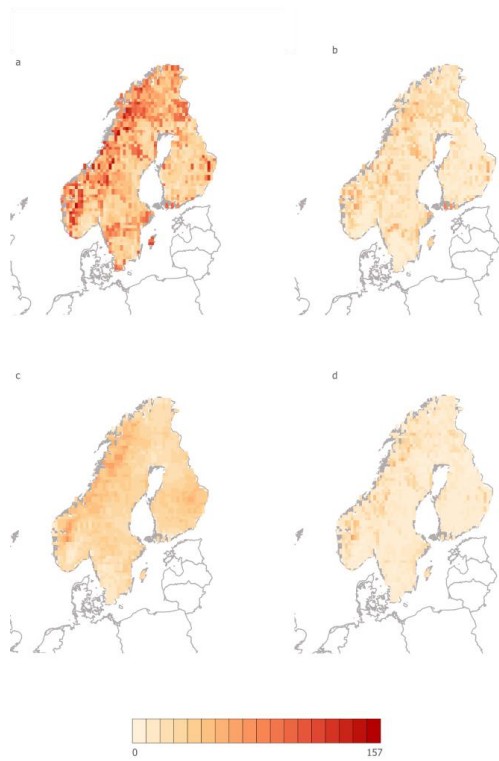

**Figure 3** Spatial distribution of compound heat-related events, in number of summer days, over the period 2000-2018
(a. Drought-wildfire, b. Heat-drought, c. Heat-wildfire, d. Drought-wildfire-heat).

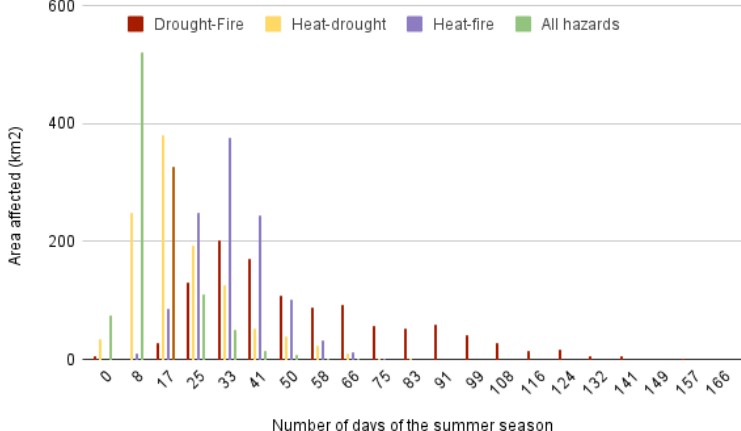

**Figure 4.** Amount of land (in km²) affected by the compound hazard combinations, in number of days, over the period
2000-2018.





We located hotspots by calculating the 90th percentile of percent of days, which can be seen in Fig. 5 below. Drought
and wildfire compound events are mainly located along the Norwegian coast (panel A). Heat and wildfire events are
also mainly located along the Norwegian coast, and along the Norwegian-Swedish northern border (panel B). Heat
and drought events are located along the Norwegian coast as well, though there are noticeably more hotspots more in-
land (panel C). There is a lot of overlap in areas where conditions are conducive to multi-hazard heat-related events.
These areas are at risk of all manner of compound events whereas most in-land regions are not at risk of any compound
events.

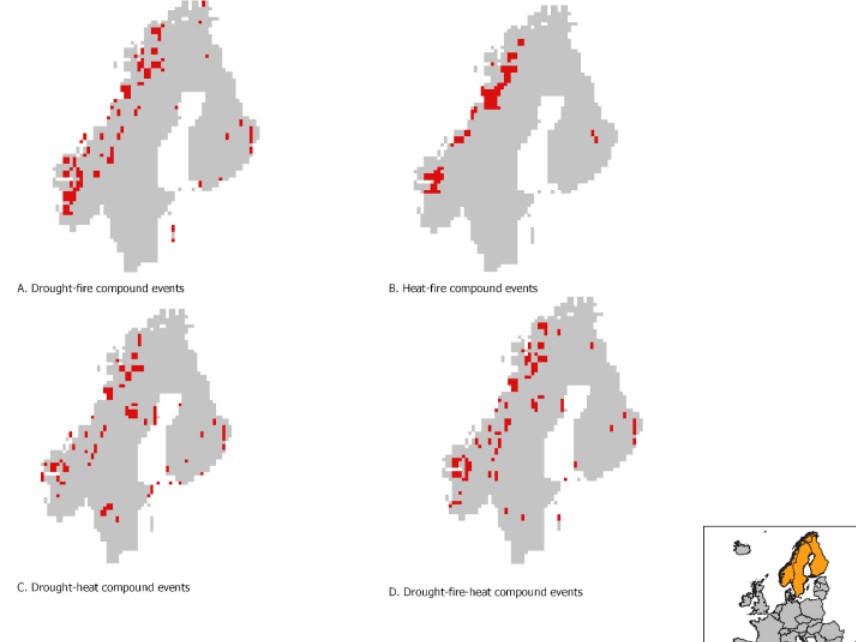



**Figure 5.** Spatial distribution of compound multi-hazard risk in Scandinavia. Figure shows the 90th percentile of
compound events over the JJA period of 2000-2018; (a. Drought-wildfire, b. Heat-wildfire, c. Drought-heat, d.
Drought-wildfire-heat).

**3.2 Land cover of 90th percentile of percent of days**
All hazard-combinations affect significantly moors and sparsely vegetated areas (Table 2). The moors are mainly
located along the northern Norwegian-Swedish border and south-western region of Norway (Fig. 6 below). Broad-
leaved forests are at high risk of all compound hazard combinations, and are found along the northern Norwegian-
Swedish border and south-western region of Norway. Coniferous forests are at quite low risk of heat-wildfire
compound events (only 5.5% of total affected area), but are at significantly higher risk of heat-drought compound
events (18.2% of total affected area). These coniferous forests are located in the south-western region of Norway,
along the northern Norwegian-Swedish border, central Sweden and in the south-eastern region of Finland. Water
bodies are at higher risk of heat-drought events than the other compound combination events. These are mainly found
in south-western Sweden (Fig. 16). Bare rocks are at high risk of all combinations of compound events.



**Table 2.** Main land cover types affected by heatwave, droughts and wildfires compound events combinations, by percent of the total area affected.

|  | Bare rocks | Broad-leaved forests | Coniferous forests | Sparsely vegetated areas | Moors and heathland | Water bodies |
|---|---|---|---|---|---|---|
| **Drought-wildfire** | 14.9 | 12.8 | 9.2 | 21.3 | 19.1 | 4.3 |
| **Heat-wildfire** | 13.6 | 15.5 | 5.5 | 16.4 | 27.3 | 2.7 |
| **Heat-drought** | 9.2 | 12.3 | 18.5 | 16.2 | 16.9 | 11.5 |
| **Drought-wildfire-heat** | 11.3 | 12.0 | 12.0 | 16.2 | 18.3 | 7.7 |






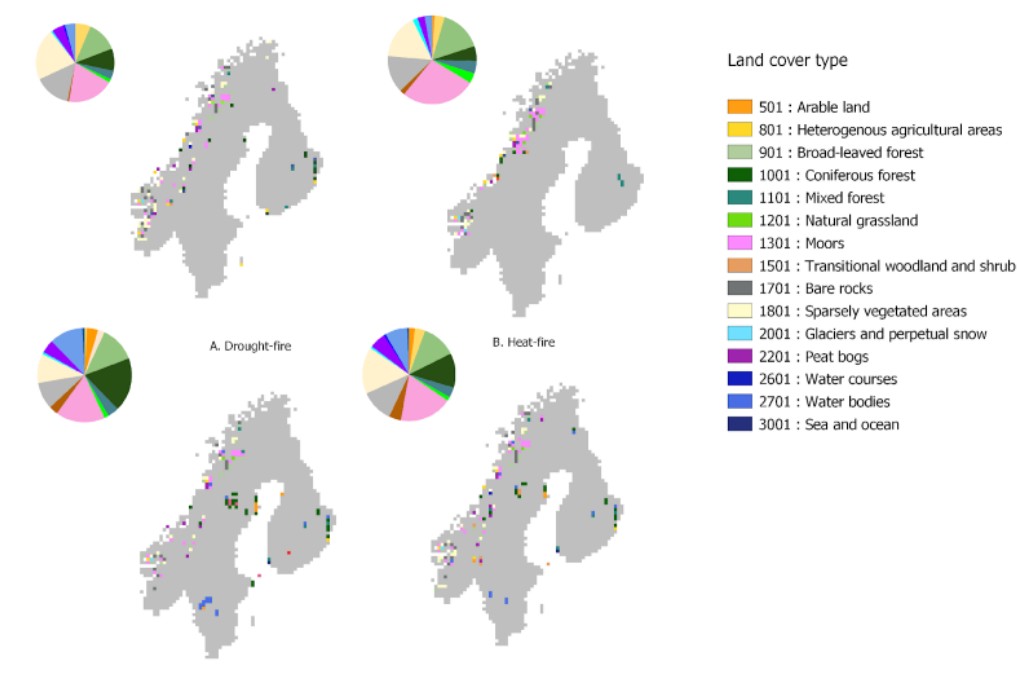


**Figure 6.** Land cover type and land cover share of areas at high risk of compound events; (a. Drought-wildfire, b.
Heat-wildfire, c. Drought-heat, d. Drought-wildfire-heat).

**3.3 Economic impact of multi-hazard on different sectors – the example of 2018**

**3.3.1 Definition of the compound event of 2018 from a Northern European perspective**

Figure 7 panel (A) shows the ERA5 maximum temperature values, averaged over the region of Finland, Norway and
Sweden, for March to September. The thick black line is the daily climatological mean over 1979-2023 and the gray
shaded area the central 90% interval over the same period. The thin black line represents daily values for 2018. Figure
7 panel (B) shows the equivalent for daily total precipitation. Overall, temperatures in spring and summer indicate
several periods of higher-than-average temperature, which in April and May coincide with periods of low
precipitation.
As described in Section 2.1, we establish an objective temporal scale for the 2018 combined heatwave and
drought by finding the scale with the smallest occurrence probability. These probabilities for all potential scales,
ranging from very short (10 days) to the full period between May and September (214 days), are shown in Fig. 8. The
central day of the respective time period is shown on the x-axis, with the event duration on the y-axis. The dots mark
the scale with the smallest probability for each duration and the X symbol the smallest probability across all scales.
Our results show that the 2018 compound heatwave and drought in Norway, Sweden and Finland occurred between
22 March and 29 July and lasted for 130 days, thus including most of the hot and dry periods seen in Fig. 7.



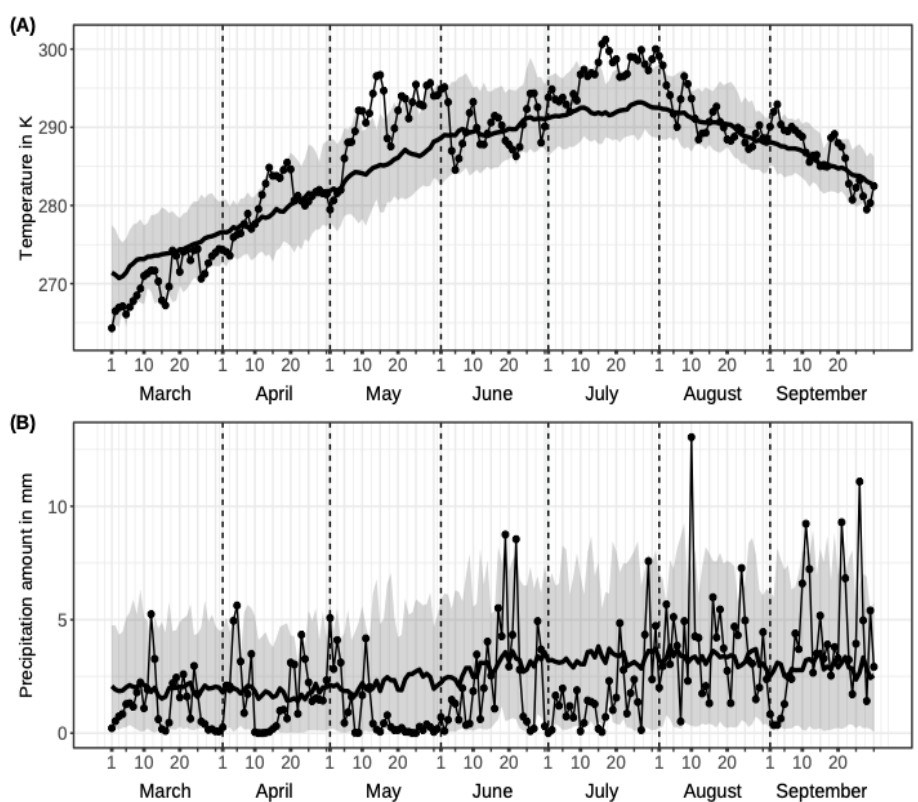


**Figure 7. Maximum temperature and precipitation**

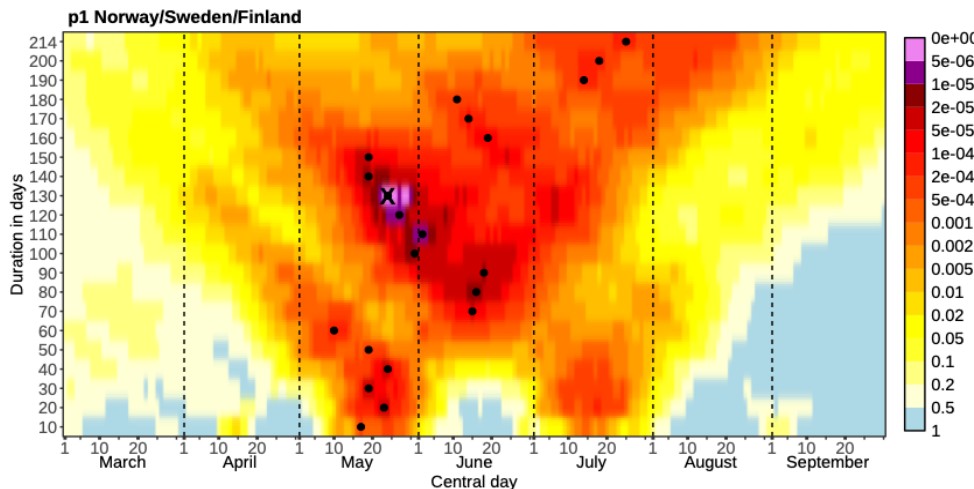


**Figure 8. Smallest probability for an event between 22 March and 29 July 2018.**





### 3.3.2 Economic impacts on the local economy

To understand the economic impacts of multiple hazards at the 2018 level, we solve the GRACE model by
incorporating the impact within the model, especially in the energy, agriculture, and forestry sectors of Scandinavian
countries for the year 2018, using estimates as shown in Table 2. The outcomes of sectoral production, prices, trade
patterns and GDP are then compared to the "business-as-usual" (BAU) case, where no hazard events occurred. The
results illustrate both the direct and indirect impact of 2018 compound events on the economy in a cross-sectoral and
cross-regional context. The impacts are evaluated for 33 European countries, other developed countries and the rest
of the world. The results presented in these subsections aggregate the impacts in Norway, Sweden and Finland.
In the Scandinavian region, the 2018 compound events contributed to an overall 0.08% drop in GDP
compared to the counterfactual scenario of BAU. It accounts for 2.23 billion NOK in 2018 value computed using 2018
GDP data collected from SSB (2024). Although this decline in GDP was moderate, it was significant enough to draw
attention and had broad impacts on the local economy.
Figure 9 depicts the changes in output by sector. Our findings reveal that the production in agriculture,
forestry, and electricity sectors all experienced negative impacts due to the direct effects of multi-hazards. Among
these sectors, the forestry sector suffered the most significant loss of 3.04%. The production of electricity decreased
by 0.50% relative to the business-as-usual case, and the agriculture sector experienced a 0.51% reduction. Meanwhile,
the lowered output in these directly impacted sectors led to an increase in the prices of the products (Fig. 10). Notably,
the domestic price of forestry goods increased by 1.64%, electricity price increased by 0.93%, and the prices of
agricultural goods rose by 0.12%.

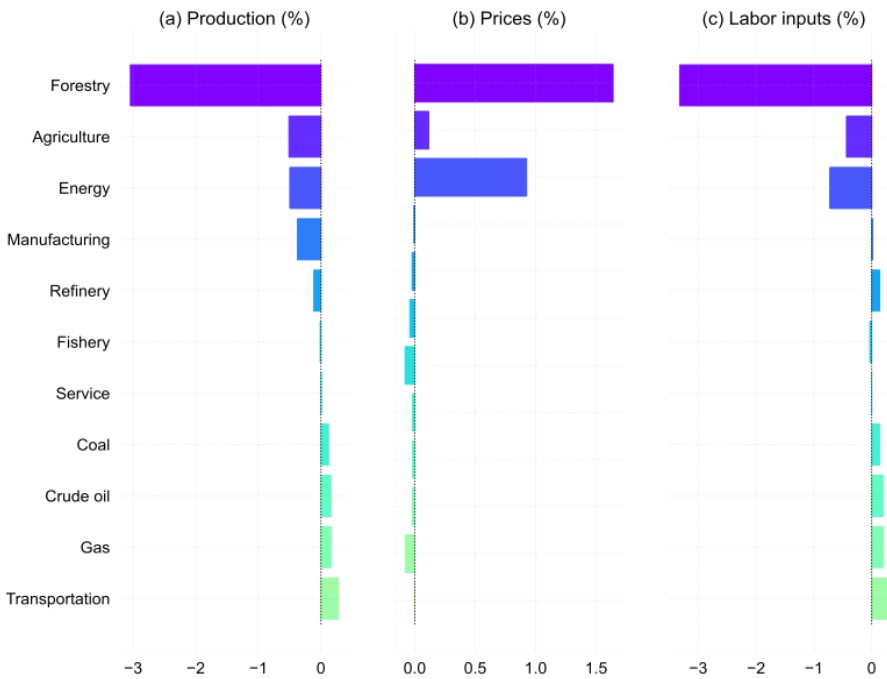

**Figure 9. Direct impacts on the domestic economy by sectors due to 2018 multi-events in Scandinavia**

The agriculture, forestry and electricity sectors are linked with other parts of the economy through their roles as
intermediate inputs. Consequently, the reduction of production in specific sectors can trigger multiplier effects. This





will result in cross-sectoral impacts beyond the initially affected sectors.  Figure 9 also demonstrates the significant
indirect impact of 2018 compound events on other sectors of the economy in the region. For instance, in panel (a) the
production of manufacturing goods had a -0.37% decline caused by the 2018 multi-hazard. It also shows that
compound events in 2018 caused a considerable indirect impact on the refined oil sector, with production dropping
by nearly -0.11% due to disturbances in energy inputs in the production process. Simultaneously, the substitution
effect resulted in an increased demand for crude oil and natural gas, boosting production in those sectors: a 0.16%
increase in crude oil production and a 0.17% increase in natural gas production. Furthermore, in panel (b), the domestic
price of fossil fuel energy moderately decreased in equilibrium due to these effects (as shown in Fig. 9). Because of
the effect on prices, the Scandinavian region would gain a comparative advantage in producing fossil fuels and
exporting supplies. This potentially led to a carbon leakage in the region.
Additionally, the decreased production in sectors directly affected by climate change in the Scandinavia
region has led to a reduction in labor demand. Consequently, there has been a decrease in labor allocated to these
sectors. As shown in Fig. 9 panel (c), labor input in the forestry sector declined by 3.32%, in the agricultural sector by
0.44%, and in the electricity sector by 0.72%. However, there has been an observed increase in labor input in other
sectors indirectly affected. While this allows for some mobility of labor within the region, the transfer of workers from
negatively impacted sectors to others does not completely compensate for the overall decrease in labor demand based
on the limitations of labor mobility. Thus, the lower labor inputs potentially lead to an increase in unemployment
within the Scandinavia region.

### 3.3.3 Economic impacts in other regions

The 2018 multi-hazard had a widespread ripple effect on the global economy, especially within Europe[2]. This is
particularly due to its significant impact on forestry goods production in the Scandinavian regions. The Scandinavian
region has an important role in exporting forestry products. Thus, the 2018 events resulted in a 29.39% reduction in
the export of forestry goods, contributing to a 0.05% drop in the trade balance, as indicated in Fig. 10a. Meanwhile,
we found that five out of eight European regions, including the British Islands, Central Europe-East, Central Europe-
North, Central Europe-West, and the Iberian countries, experienced a decline in their trade balance. These regions are
important trading partners of Scandinavian forestry products which highlights the widespread economic impact of
2018 multi-hazards across Europe.
Despite having a strong forestry sector, the Baltic region is projected to see a 0.03% decrease in its trade
balance. This decline is largely due to the dominant position of Scandinavian forestry products in the global market.
The reduced supply of forestry goods from Scandinavia could not fulfill the global demand and increased prices of
wood products worldwide. As illustrated in Fig. 10 panel (b), wood products from the Baltic states have experienced
a 0.39% price increase, the second highest price increase after the Scandinavian region. The large surge in prices
created a comparative disadvantage for Baltic forestry products in the global trade market, making them less
competitive compared to alternatives. Consequently, this explains the negative ripple effect on the trade balance
volume in the Baltic states.

---

[2] The results on the country level impacts are aggregated into 8 sub-regions within Europe. Scandinavia includes
Norway, Sweden and Finland. Baltic States include Estonia, Latvia, Lithuania. The British Isles include Ireland and
the United Kingdom. Eastern Central Europe includes Denmark, Germany, Netherlands, Belgium and Luxembourg.
Eastern Central Europe includes the Czech Republic, Hungary, Poland, Slovakia, Slovenia, Albania, Bulgaria,
Belarus, Croatia, Romania, and Ukraine. Iberian Peninsula includes Spain and Portugal. Southern Central Europe
includes Cyprus, Greece, and Italy. Western Central Europe includes Malta, Austria, and France.



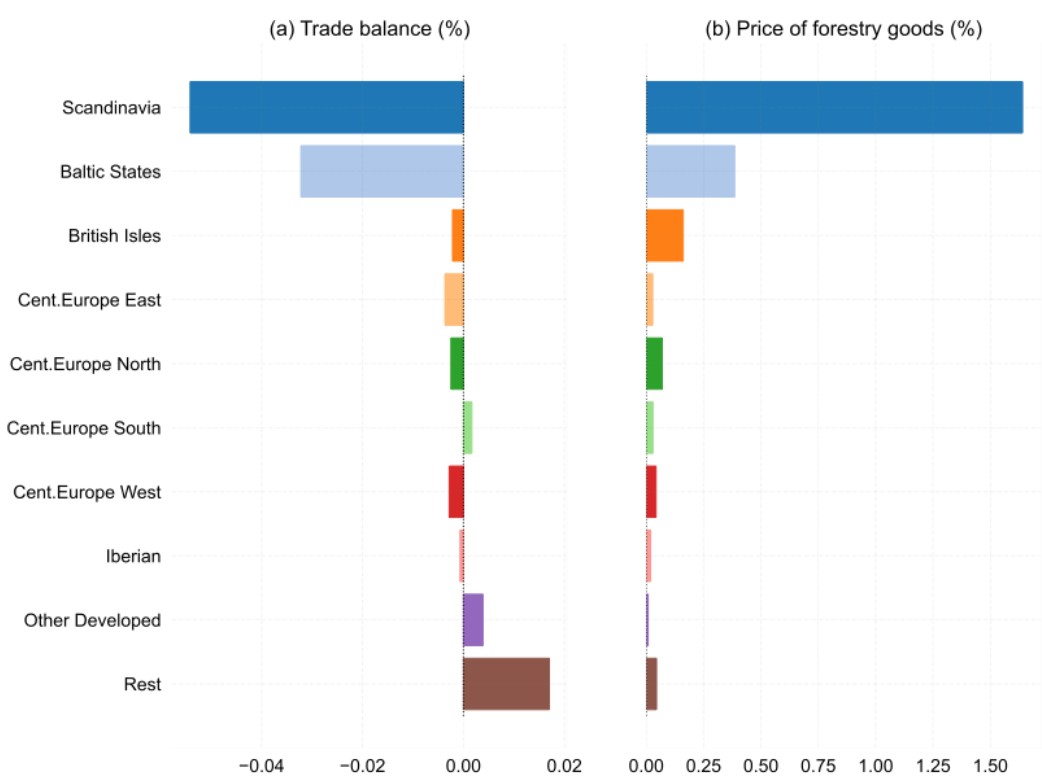


**Figure 10.** Abroad economic impacts due to 2018 multi-events

Interestingly, our findings indicate that the 2018 disturbances in Scandinavia have stimulated the export of forestry
products from other developed countries and the various regions in the rest of the world. These areas have experienced
more moderate price increases (as shown in Fig. 10 panel (b)), leading to an implied comparative advantage. It
motivated production and export from these remote countries to meet the global demand for forestry products.
According to FAOSTAT (2024), the major exporters of forestry products aside from European countries, include
developed countries such as the United States, Canada and Russia. The list also extends to other countries, including
China and Brazil, among others. These countries also increased their market presence in the forestry sector and thus
compensated for the reduced supply from the Scandinavian region.
Ultimately, the market effects and trade effects transform the direct, sector-specific impacts into broader
cross-sectoral and cross-regional impacts. These cumulative effects contribute to the impact on the GDP of each
region. Figure 11 presents the isolated impact on GDP due to the 2018 events in 33 European countries. As shown in
Fig. 11, countries in the Baltic states, British Isles, and Central Europe-East have experienced GDP losses caused by
2018 compound events, mainly driven by inter-regional trade effects. In contrast, countries in Northern Central
Europe, Southern Central Europe, and the Iberian regions have seen GDP growth during the period. The GDP growth
in these regions is the result of the combined effects of changes in internal markets or trade patterns. Particularly, Fig.
10 shows that Southern Central Europe benefited from the remote impact in the Scandinavia region with an increase
in the trade balance due to rising prices of forestry goods, leading to a positive GDP growth as shown in Fig.11.




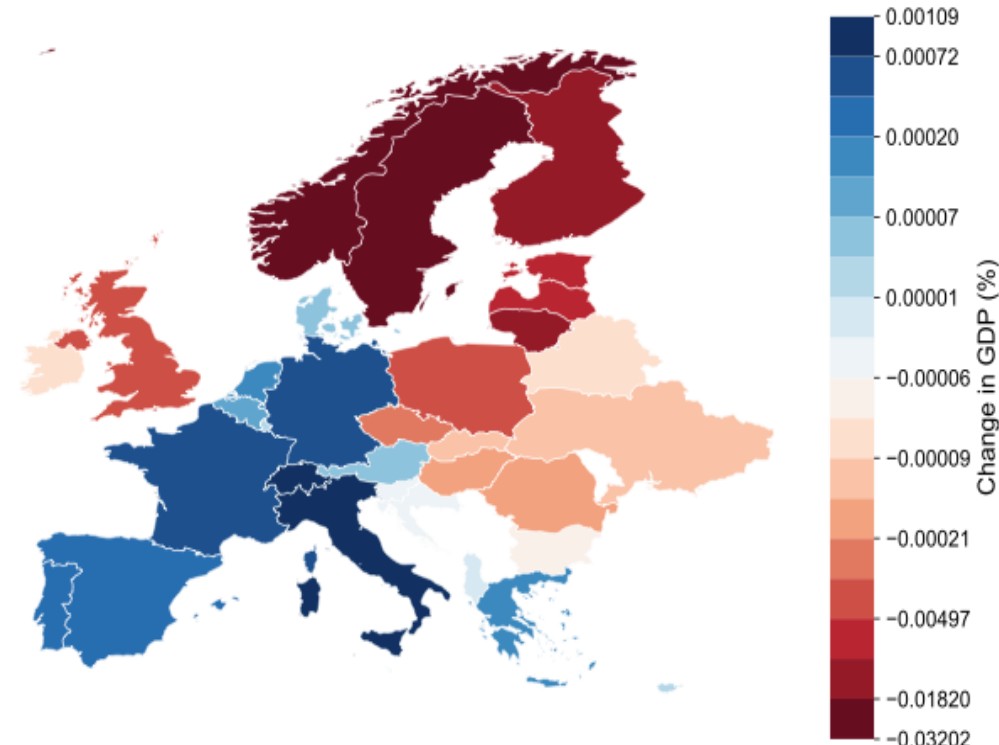


**Figure 11.** Economic impacts of 2018 compound events: GDP impacts in 33 European countries


## 4 Discussion

This study aims to better understand the impacts and occurrences of multi-hazard events in summer (such as compounding heatwave, drought and wildfire) in Scandinavia from a multidisciplinary perspective. High risk drought-wildfire events occur twice as often as heat-wildfire, and heat-drought events, with spatial patterns ranging primarily along the Norwegian coast, which is in accord with the results from Sutanto et al. (2020). Combinations of wildfire-hazard events primarily occur in Norway, with fewer occurrences in Finland, where several factors such as natural fire breaks and an extensive road network help maintain the fires small and at a low-intensity (Fernandez-Anez et al., 2021). Forest management in Finland is as such that a large majority of the biomass is removed during harvesting, decreasing the amount of available fuel (Fernandez-Anez et al., 2021), although it has been noted that an increase in prescribed burning would be beneficial in order to increase forest biodiversity in the country (Lindberg et al., 2020).

Drought-hazard events in Sweden appear to occur in the southern and central regions, where Teutschbein et al. (2022) found that southern catchments experienced more severe streamflow droughts than northern ones. Teutschbein et al. (2022) identified a wetting trend in Sweden during the winter months, with a minor drying trend during the spring and summer, which suggests that drought management measures should be put into place at a regional scale, where regional differences in climate might occur. Blauhut et al. (2022) also mention the urgency of an European drought governance approach in the form of a general framework permitting flexible regional management strategies.



All multi-hazard combinations affect significantly moors and heathlands, mainly located along the northern
Norwegian-Swedish border and south-western region of Norway (Fig. 5). Dead heather specimens in low humidity
air were found by Log et al. (2017) to dry at a surprisingly fast rate, showing they were prone to fire "within two days
during wintertime and well within one day in warm weather".  During the winter of 2014, after a 3-week period with
no precipitation registered by the Norwegian Meteorological Institute and relatively windy weather, wildfires burned
in 2014 a total 35 km² surface area of heathlands (Log et al., 2017). Prescribed burnings had not been performed in
the area over the last 50 years, and resulted in an accumulation of dead heather and thus vegetation susceptible to
drying and wildfires (Log et al., 2017). Unmanaged heathlands thus pose a fire risk in dry and windy weather, and
would benefit from mitigation management measures, especially with fire and drought frequency expected to increase
in boreal ecosystems.
The main land cover types at risk of heat-related multi-hazards in Scandinavia are vegetated (broad-leaved
forests, coniferous forest, sparsely vegetated areas). In multi-hazard hotspots, namely along the northern Norwegian-
Swedish border,  south-western region of Norway,  central Sweden and in the south-eastern region of Finland, forest
management mitigation measures could be implemented to decrease this risk. Certain zones at high risk of multi-
hazards have actually seen an expansion of a specific land cover (for example Norwegian broad-leaved forests). These
regions would benefit from implementing suitable adaptation measures, to decrease the vulnerability of such areas.
Not anticipating possible hazards could result in economic losses if a hazard does occur, for example California's
timber production was severely affected by a forest die-off event attributed to the 2012-2015 drought (Sleeter et al.,
2018). As Sweden's and Finland's economies rely on wood products production and export, it is important to ensure
forested areas are adapted to droughts, wildfires and heat waves, particularly when anthropogenic climate change is
predicted to intensify fire and drought frequency in boreal ecosystems (Girardin et al., 2010 ; IPCC, 2021). Especially,
our economic assessment of the impact of 2018 multi-hazards reveals a varying and wide-spreading result across
sectors and regions, particularly in Europe. Consistent with Beillouin et al. (2020), Bakke et al. (2020) and Gustafsson
et al. (2019), our results include reduced agriculture, energy and forestry output in the Scandinavian region as the
direct impacts. The sectoral-specific impacts also transfer to other sectors in the Scandinavian economy. For example,
we find a decrease in manufacturing production caused by reduced intermediate inputs of agriculture, energy and
forestry goods. At the same time, we also find an increase in the production of oil and gas due to the substitution effect
of less electricity production. Furthermore, the compound event of 2018 also affected the trade of forestry goods
because of the vital role of Scandinavia in the international wood market. This led to a moderate yet widespread effect
on GDP losses, affecting not only the Scandinavian region but also trading patterns, particularly in Europe.  Sparsely
vegetated areas could also benefit from monitoring drought or fire risk in the area. Human activity is responsible for
more than 80% of wildfires in Europe, with data suggesting that about 60% of fires are started deliberately (EEA,
2020), and human-induced fires spread faster than lightning-induced fires (Hanston et al., 2020). Awareness
campaigns to reduce the risk of ignition in areas where vegetation is vulnerable to drought or fire could be carried out
by regional governments. Bare rocks are at such a high risk of heat-related multi-hazards due to Sutanto et al.'s (2020)
calculations being based on atmospheric data and soil moisture. Bare rocks have low moisture content compared to
vegetation, which could explain that they cross the soil moisture drought threshold when only looking at soil moisture.
**Limitations and outlook**
This study has potential limitations for risk mapping, evaluating impacts, among others. The multi-hazard risk maps
were put together using atmospheric data originally used for a heatwave, drought and wildfire risk analysis of
continental Europe, which resulted in coarser resolution when cropping to the Scandinavian region. The aim of this
study was not to generate new data but to use this previous research to produce multi-hazard risk maps of the selected
regions. Due to the scope of this study, the land cover datasets were retrieved from Copernicus' Land Monitoring
Service instead of national land cover datasets. This helped keep the land cover analysis consistent for all three
countries included in the study, but also rendered a coarse land cover map of Scandinavia.
When assessing the economic impacts of 2018 multi-hazard, our approach also faces certain limitations. First,
there is a lack of robust models capable of evaluating the physical impact of multi-hazards on energy and agriculture



production. In this research, we employ historical data to estimate the direct impacts of climate change-relevant multi-
hazards. Employing past events as a reference point for extreme scenarios could potentially lead to underestimations.
Second, our current assessment does not include climate change impacts in regions outside the primary area of study,
which may have a significant effect on the socioeconomic impact in the Scandinavia country. This highlights the need
for more comprehensive data collection and modeling to assess the direct and indirect impact of multi-hazards in a
broader scope.
Three main extensions of this study could be potentially considered. Firstly, since drought risk was calculated
by Sutanto et al. (2020) by looking at soil moisture data, which specifies soil moisture drought, this study could also
be expanded to consider another type of drought (such as hydrological or meteorological drought) when calculating
drought risk. Blauhut et al. (2022) recommend, to improve drought risk management, to look at different types of
drought, which use different indicators and impact different sectors. For example, a study done by Asner et al. (2015)
assessed the 2012-2015 drought in California by looking at forest canopy loss, which displayed a broader range of
drought-affected forests than was seen with visual mapping approaches. Secondly, multi-hazard risk maps were
generated using past atmospheric data, from 2000 to 2018; an extension of this study could be made by building multi-
hazard risk maps on future climate scenarios. Various studies have looked at future drought risk in Europe, such as
research conducted by Roudier et al. (2015) and Spinoni et al. (2017), which could provide geospatial data to map
future drought risk. Third, we suggest a close investigation into how the stock and productivity of forestry were
affected by the 2018 multi-hazards using land surface models, for example, Community Land Model (CLM)
(Lawrence et al., 2019). The approach would provide a more accurate assessment of the losses in the forestry sector
and also help to refine its spill-over effect on the broader economy. We also recommend extending similar sectoral-
specific models for agriculture and energy sectors to capture the full scope of 2018 multi-hazard impacts.
Forest management and adaptation measures are crucial to reducing the risk of heat-related multi-hazards in
vulnerable vegetated areas of Scandinavia, particularly in multi-hazard hotspots like the Norwegian-Swedish border,
as droughts and wildfires, intensified by climate change, could severely impact timber production and regional
economies reliant on wood exports. The findings of this study can provide guidance for policy makers regarding forest
management in Scandinavia in the current context of anthropogenic climate change. By highlighting the
interconnectedness of heat-related events, we aim to emphasize the importance of anticipating these hazards,
particularly droughts and wildfires, ultimately mitigating their impacts on the environment and the economy.

## 5 Conclusions

To better understand the interplay of multi-hazard risk of heatwaves, droughts and wildfires in a multi-sectoral context
and to improve disaster risk management in a multi-hazard setting, we assess the occurrence of these hazards using a
spatial analysis of compound heatwave, droughts and wildfires events from 2000 to 2018 in Scandinavia. Our results
show that high risk drought-wildfire events occur twice as often as heat-wildfire, and heat-drought events, with
occurrences up to 166 days of the summer seasons between 2000-2018. Furthermore, our analysis suggests that
hotspots for compound drought, heat, and wildfire events in Scandinavia are primarily concentrated along the
Norwegian coast and the northern Norwegian-Swedish border, with significant overlap in areas prone to all multi-
hazard combinations, while inland regions are generally not at risk. When looking at the economic impacts of the
2018 compound multi-hazard events, an 0.08% GDP drop in Scandinavia was observed, primarily impacting the
forestry sector, which saw a 3.04% decline, alongside cross-sectoral effects and increased prices in agriculture,
forestry, and electricity. Furthermore, the same event led to a 29.39% reduction in Scandinavian forestry exports,
causing a ripple effect across Europe, with trade balance declines in five European regions and a 0.05% overall drop
in the trade balance due to the disruption in the global supply of forestry products. Effective forest management and
adaptation are key to reducing the risk of heat-related multi-hazards in vulnerable Scandinavian regions, especially
along the Norwegian-Swedish border, where droughts and wildfires, exacerbated by climate change, threaten timber
production and regional economies. This study offers guidance for policymakers on mitigating these interconnected
hazards to protect both the environment and the economy.



*Data availability*. ERA5 reanalysis data is openly available from the Copernicus Climate Change Service (C3S)
Climate Data Store (CDS) at https://cds.climate.copernicus.eu/datasets/reanalysis-era5-single-levels.
CORINE Land Cover data (CLC) is openly available from the Copernicus Land Monitoring Service at
https://land.copernicus.eu/en/products/corine-land-cover.

*Author contributions*.
Author contributions follow the CRediT authorship categories.

**Gwendoline Ducros**: Conceptualization, Writing - Original Draft, Data Curation. **Timothy Tiggeloven**:
Conceptualization, Writing - Review & Editing, Supervision. **Lin Ma**: Software, Writing - Original Draft, Data
Curation. **Anne Sophie Daloz**: Conceptualization, Writing - Review & Editing. **Nina Schuhen:** Investigation,
Software, Writing - Original Draft. **Marleen de Ruiter**: Conceptualization, Writing - Review & Editing, Supervision.

*Competing interests*. The contact author has declared that none of the authors has any competing interests

*Acknowledgements*.
T.T., A.S.D., L.M., and M.d.R. were supported by the European Union's Horizon 2020 funded project MYRIAD-EU
(Grant 101003276). MCR also received support from the Netherlands Organisation for Scientific Research (NWO)
(VENI; grant no. VI.Veni.222.169). Furthermore, this study was partially supported by the ACRoBEAR project, a
project under the Belmont Forum Collaborative Research Action on Climate, Environment and Health.

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
