# Peer review of "Multi-hazards in Scandinavia: Impacts and risks from compound"

_EGUsphere, 2024_

## Author Response (AR1)

**Multi-hazards in Scandinavia - Response**

**Comment #1**

The manuscript analyses compound droughts, heatwaves and wildfires hazards in Scandinavia and assesses direct and indirect economic impacts to hydropower, agriculture and other socio-economic indicators. The topic of multi-hazards and multi-risks is extremely interesting and deserves high attention from the researches. Therefore I congratulate with the authors for giving their contribution to this topic. In my opinion the manuscript is well written and organized, and the methodology is well described. There are in my opinion two main issues to be resolved by authors to improve the quality of the manuscript.

(1) By reading the introduction and the methods the general impression is that this work combines several existing data and methods (e.g., GCE model GRACE, ERA5 data) without spending much words on the actual contibution of the authors and how the applied methods in the selected context provide an innovative perspective on the multi-hazard theme. In other words, what the work reveals that could not be captured, or well interpreted, or understood, without the multi-hazard analysis? What is the added value? And how these insights help in adaptation? (adaptation is mentioned in the abstract without much results or discussion on that in the body of text).

Thank you for the comment. We added the following paragraph in the Discussion section:

"Through the integration of different methodologies, including spatial analysis and the GRACE model, this research assesses the economic impacts of the 2018 heatwave, drought, and wildfires in Scandinavia. However, beyond simply applying these tools, our key contribution lies in the explicit multi-hazard perspective, as hazards rarely occur in isolation. Single climate events in Scandinavia, such as increased temperature, have been shown to enhance production in certain sectors. For instance, moderate warming may extend the growing season and can benefit the agriculture and forestry sectors (Maracchi et al, 2005). However, our study highlights that when extreme heat co-occurs with drought and fire, the overall economic impact is widespread across multiple sectors. This underscores the importance of analyzing multi-hazard dynamics rather than assuming independent effects."

Additionally, we agree with the review that the adaptation part does not have much body in the main text, so this concept will be removed at places where it does not have added value.

(2) In the results the description of some of the figures/plots is missing. As if some of the figures does not contribute to the 'story' of the paper. I would suggest to clearly describe and discuss the provided images. Particularly Fig. 4 (not described and poorly readable) and Fig. 8.

Thank you for the comment, Fig. 4 will be updated to enhance its readability, and we will add a description. For Fig. 8, we also added "This duration is used as a proportional weight while assessing the annual economic impact of the 2018 multi-hazard"

**Minor issues:**

In table 2 Bare rocks are listed as 'affected' by heatwaves, droughts, fires and their combination. Why? are bare rocks vulnerable? I think a bit of discussion and comment on this would be precious for the reader.

Thank you for the comment, we will add a discussion point as to the relevance of mentioning "bare rocks" affected by multi-hazards.

Figure 9. Caption. It seems that also indirect impacts are shown, not only direct.

We appreciate the referee's comment. That is true. The caption of Figure 9 will be revised as "Direct and indirect impacts on the domestic economy by sectors due to 2018 multi-events in Scandinavia".

There are some typos, e.g. reference to Fig.16 that does not exist (I. 273), please check the text carefully.

We will remove the typos and will go over the manuscript another time in detail to filter potential other typos.

**Comment #2**

Thank you for the insightful article on multi-hazard impact assessment in Scandinavian countries, focusing on multi-hazard events that occurred in 2018. The study addresses a crucial topic and comprehensively covers multiple dimensions of risk—hazard, exposure, and impact. However, I would like to suggest a few improvements to enhance the transparency and robustness of the analysis. Specifically, it would be beneficial to include details on the models used, their parameters, validation metrics, and relevant assumptions. Providing this information either in the main manuscript or the supplementary materials would allow for a more thorough evaluation of the analysis's robustness and completeness.

While the overall scope of the article is commendable, there are significant gaps in defining a clear research objective. In its current form, the objective appears somewhat vague. For instance, the phrase "assessing the occurrence" of multi-hazard events is unclear. Does this refer to calculating the probability of co-occurrence, conditional probabilities, or something else? Clarifying this would help readers better understand the study's goals and focus.

Methods: The methodology section introduces a combination of variables from reanalysis data (ERA5), multi-hazard occurrences, and impact models. However, the connection between these elements is not sufficiently clear. This lack of coherence makes it difficult to follow how the methodology aligns with the stated objectives. It also reflects in the results and discussion sections. For example, it is unclear what is the role of the hotspots identified in Figure 8 in the multi-hazard impact assessment. Providing a more structured explanation of the methodology and how each step contributes to the overall analysis would significantly improve the manuscript.

We thank the reviewer for the comprehensive thoughts on improving the manuscript, specifically on the transparency and robustness of the analysis. We agree that the manuscript should be enriched with additional information on the models used and refining the research objective more. We will address the specific points listed below and we will integrate the comments and feedback into the revised manuscript.

- Abstract: While the relevance of multi-hazard risk assessment is well-highlighted, the abstract would benefit from a more explicit statement of the study's objective and the key takeaways from the results.
  - We appreciate the reviewer's comment and we have identified one key sentence on the objective of the manuscript and another summarizing the key findings.
  - "To better understand the interplay of multi-hazard risk of heatwaves, droughts and wildfires in a multi-sectoral context and to improve disaster risk management in a multi-hazard setting, we assess the occurrence of these hazards using a spatial analysis of compound heatwave, drought and wildfire events in Scandinavia."

"This research shows the importance of ripple effects of multi-hazards, specifically compound heatwave, drought and wildfire, and that forest management and a better understanding of their

direct and indirect societal impacts are vital to reducing the risks of heat-related multi-hazards in vulnerable areas."

• Introduction: The literature review is strong and provides good context for multi-hazard analysis. However, the study's specific objective remains vague. Additionally, it would be useful to include references to previous research on hazard impacts in Scandinavia and discuss the contexts in which the methods used in this study were developed. How relevant is the macro-economic model to the specific case study?

We appreciate the reviewer's comment and we will make the study's specific objective more clear with better defined statements in the introduction. We will also include more previous research and provide more context of the study area.

Key objective sentence: "As the probability of occurrence of similar events of the 2018 multi-hazard in Scandinavia is increasing with climate change (REF), it is crucial to better understand the interplay of multi-hazard risk of heatwaves, droughts and wildfires in a multi-sectoral context with economic ripple effects. In this paper, we assess the occurrence of these hazards using a spatial analysis of compound drought, wildfire, and heatwave events from 2000 to 2018 in Scandinavia (here Finland, Norway and Sweden), and assess the direct and indirect impacts through a macro-economic model. Secondly, to assess their potential direct and indirect economic impacts we use the global Computable general equilibrium (CGE) model GRACE (Global Responses to Anthropogenic Changes in the Environment) and the 2018 heatwave-drought period as a baseline."

Relevancy model: "The macroeconomic model provides a comprehensive and regionally relevant assessment of how sector-specific shocks from the 2018 multi-hazard events in Scandinavia propagated through the economy, revealing both direct and indirect impacts."

**Added references:**

Spinoni, J., Vogt, J. V., Naumann, G., Barbosa, P., & Dosio, A. (2018). Will drought events become more frequent and severe in Europe?.

Berghald, S., Mayer, S., & Bohlinger, P. (2024). Revealing trends in extreme heatwave intensity: applying the UNSEEN approach to Nordic countries. Environmental Research Letters, 19(3), 034026.

Spensberger, C., Madonna, E., Boettcher, M., Grams, C. M., Papritz, L., Quinting, J. F., ... & Zschenderlein, P. (2020). Dynamics of concurrent and sequential Central European and Scandinavian heatwaves. Quarterly Journal of the Royal Meteorological Society, 146(732), 2998-3013.

Wilcke, R. A. I., Kjellström, E., Lin, C., Matei, D., Moberg, A., & Tyrlis, E. (2020). The extremely warm summer of 2018 in Sweden–set in a historical context. Earth System Dynamics, 11(4), 1107-1121.

• Methods: The explanation of the methodology (lines 82–92) is fragmented, making it difficult to follow. Consider providing an overview of the rationale behind each methodological step before

- explaining the individual steps. Additionally, the flowchart could be improved by clearly separating data sources (e.g., ERA5) from methods (e.g., copula modeling). We appreciate the reviewer's comment and will add an overview of the rationale between each methodological step. We will look at how we can make the distinction in the flowchart clearer for the reader.
- Lines 117–121: The definition of multi-hazard event probabilities is hard to follow. Could you provide a couple of equations to clarify this? Specifically, what does the condition "greater than or less than the 2018 value" mean, and what time periods are being considered?
   We have expanded this section to explain the method, through which we derive the optimal event definition, in more detail and we hope that this is now much clearer.
- What metrics were used to select the best-fit copula for modelling the coupled marginal distributions of precipitation and temperature?
   We have added a sentence stating "In this case, we found that the symmetrical Frank copula is the best fit for our data set, by minimizing the Akaike Information Criteria across several copula families (as implemented in the R package VineCopula; Nagler et al., 2024). The copula parameters are determined using the inverse of the Kendall rank correlation coefficient."
- It is unclear how the copula definition of the 2018 event connects to subsequent steps in the
  methodology. If I understand correctly, the later steps do not consider the probability of
  (temperature and precipitation) exceedance. Could this connection be clarified?
   We have added a sentence stating that the event definition is used as a proportional weight for
  assessing the economic impact of the 2018 multi-hazard event..
- More information on the econometric models is needed. How complete are the confounding variables? It appears that key economic and spatial variables may be missing. Providing details on regression model performance and stepwise variable selection in the Supplement/Appendix would enhance transparency.
  - We appreciate the reviewer's comment. We have enclosed the estimation results in the Appendix. The variables for selection include variables [dT, dP, dT^2, dP^2, TdT, PdP, dTdP]. Meanwhile, for all the estimations, we included year effects and country-fixed effect when estimating the imposed response function for agriculture production. We included both year and month effects for the estimation for the energy production.
- What are the uncertainties in the macro-economic model? How well do the model's predictions align with observed GDP changes in the affected regions?
   We appreciate the reviewer's comment. The CGE model used in this study, like all macroeconomic models, has uncertainties that can be summarized in the following aspects. First, the GRACE model relies on input-output tables, which represent fixed sectoral interdependencies. However, during hazards, these relationships can change dynamically due to shifts in production structures, substitution effects, and market adjustments. In this study, we focus on the yearly impact of the 2018 multi-hazard. Given that major structural adjustments within a single year are rare, the model's static representation of sectoral interdependencies is a reasonable assumption for short-term analysis.

Second, uncertainty arises from the GRACE model's parameters. The CGE model relies on key parameters, such as substitution elasticities between capital, labor, and natural resources. These parameters are based on values from the literature and previous studies, as a common practice in CGE modeling. The uncertainties of model outcomes to these elasticities exist. However we use baseline dataset for model calibration, to ensures that the model replicates key features of the regional economy as accurately as possible in the base year.

Third, there may be uncertainty in the damage function estimation. The economic impacts of heat, drought, and fires are incorporated through productivity loss functions. These functions are necessarily simplified representations of complex biophysical and economic processes. While they provide useful estimates, they introduce uncertainty in quantifying the exact magnitude of sectoral impacts.

Regarding the comparison with observed economic changes, particularly GDP, we find it is challenging as real-world GDP outcomes are influenced by multiple overlapping factors beyond the specific multi-hazard event in this study. Observed changes in GDP, production, and prices reflect market equilibrium adjustments that include autonomous adaptation to multiple disturbances. Our study isolates the impact of the 2018 multi-hazard event, making direct GDP comparisons complex.

While the change in the GDP level may not provide a clear validation of the model, we find that the growth of real GDP in Scandinavian region did slow down from 2.4% in 2017 to 1.4% in 2018, based on the World Bank (2025) dataset. The reduced growth rate potentially reflects the economic consequences of the 2018 events on the aggregate level.

Furthermore, we also find consistent evidence in observed price trends. According to Statistics Sweden (2025) and the Swedish Forest Agency (2025), the average real price of sawlogs and pulpwood increased by 5.7% and 11.5%, respectively, in 2018 (real price adjustments based on CPI from Statistics Sweden). In Norway, the real price of timber products increased by 12% in 2018, based on data from Statistics Norway (2025). In Finland, the real price of timber products increased by 4% in 2018 (Statistics Finland, 2025). Similarly, the real price of electricity grows notably by 50% in Norway, 12% in Sweden and 5% in Finland.

Our study finds a consistent directional change in the prices of forestry products and electricity. However, the magnitudes observed in empirical data are somewhat larger than our CGE model evaluation. Several factors may explain this. 1), the CGE model provides average changes across the agriculture, energy, and forestry sectors for Scandinavia as a whole, potentially smoothing out regional price variations. 2) the empirical price trajectory may reflect broader market dynamics, where supply-chain disruptions and energy market responses extended beyond Scandinavia, amplifying price changes beyond what the model captures at a regional level. 3) the CGE model represents an equilibrium state based on rational economic behaviour, but in real-world markets, short-term speculative behaviour can drive prices higher. For instance, following a supply shock due to heat, drought and fires, traders and businesses might anticipate

further shortages of production of energy, agriculture and energy products, leading to price spikes that overshoot equilibrium predictions. Thus, while GDP comparisons remain challenging due to confounding factors, the model successfully captures the qualitative patterns of price adjustments observed in the market.

Results and Discussion: The discussion on the contrast between single- and multi-hazard modeling in the introduction is an interesting point. It would be valuable to illustrate how modeling multi-hazards adds insights compared to single-hazard models. Does it bring the predictions closer to ground truth with respect to impact assessments?
 We thank the reviewer for this comment. We agree and will more explicitly illustrate, argue and discuss how modelling multi-hazards adds insights compared to single-hazard analyses.
 "By moving towards assessments that include compounding effects, modeling multi-hazards provides a more realistic representation of systemic risks, offering insights into indirect impacts that single-hazard models often overlook, and thereby improving the relevance of impact assessments."

The manuscript does a good job of highlighting the limitations. However, it remains unclear how these limitations quantitatively affect the estimates. How far do these uncertainties impact the impact estimates? Incorporating quantitative or qualitative validation would significantly strengthen the manuscript's credibility and help readers assess the reliability of the findings.

Thank you for the comment. We intend to strengthen the credibility of findings by incorporating qualitative validation, particularly focused on the observed price trends, as discussed in the question of "What are the uncertainties in the macro-economic model? How well do the model's predictions align with observed GDP changes in the affected regions?"

Meanwhile, we also added the following paragraphs in the Discussion section.

"To validate the findings from the economic impact assessment, we compared our results with the observed price trends of electricity and forestry products. Since agricultural goods in Scandinavia tend to be more regulated (through subsidies and state-controlled food reserves) than other goods, the agricultural prices did not respond so much to extreme weather events. Collecting CPI from Statistics of Sweden (2025) and roundwood prices from the Swedish Forest Agency (2025), we find in 2018 the average real price of sawlogs and pulpwood increased by 6% and 12% respectively. In Norway, the real price of timber products increased by 13% in 2018, based on data from Statistics Norway (2025). In Finland, the real price of timber products increased by 4% in 2018, using data from the Natural Resources Institute Finland (2025). Similarly, the real price of electricity also surged, growing notably by 50% in Norway (Statistics Norway, 2024), 12% in Sweden (Statistics Sweden, 2025) and 5% in Finland (Statistics Finland, 2025). Our study finds moderate yet consistent directional change in the prices of forestry products and electricity as shown in Figure 9. The larger price changes observed in empirical data can be attributed to the broader market dynamics caused by the 2018 multihazard. At the same

time, in the real-world market, short-term speculative behavior can drive prices higher as traders and businesses anticipate future production disruptions, a feature not captured in the GRACE model. We find it remains challenging to validate impacts on GDP levels using empirical observations. This is because our study isolates the effect of the 2018 events, whereas the observed values in the national accounts are influenced by various factors beyond the specific multi-hazard events in this study. However, we find that the growth of real GDP in Scandinavia showed down from 2.4% in 2017 to 1.4% in 2018, based on the World Bank (2025). The reduced growth rate potentially reflects the extensive economic consequences of the 2018 events at the aggregate level."

In the end, we also added the correspondent references.